# SAND: Smooth Imputation Of Sparse And Noisy Functional Data With Transformer Networks

**Ju-Sheng Hong**[*]
Department of Statistics
University of California Davis
Davis, CA 95616
jsdhong@ucdavis.edu

**Junwen Yao**
Department of Statistics
University of California Davis
Davis, CA
jwyao@ucdavis.edu

**Jonas Mueller**
Cleanlab
Cambridge, MA
jonaswmueller@gmail.com

**Jane-Ling Wang**
Department of Statistics
University of California Davis
Davis, CA
janelwang@ucdavis.edu

## Abstract

Although the transformer architecture has come to dominate other models for text and image data, its application to irregularly-spaced longitudinal data has been limited. We introduce a variant of the transformer that enables it to more smoothly impute such functional data. We augment the vanilla transformer with a simple module we call SAND (**s**elf-**a**tte**n**tion on **d**erivatives), which naturally encourages smoothness by modeling the sub-derivative of the imputed curve. On the theoretical front, we prove the number of hidden nodes required by a network with SAND to achieve an $\epsilon$ prediction error bound for functional imputation. Extensive experiments over various types of functional data demonstrate that transformers with SAND produce better imputations than both their standard counterparts as well as transformers augmented with alternative approaches to encode the inductive bias of smoothness. SAND also outperforms standard statistical methods for functional imputation like kernel smoothing and PACE.

## 1 Introduction

Functional data analysis (FDA) offers a framework for analyzing complex data sampled from random functions or curves. Such data are encountered in many applications including air pollution studies, fMRI scans, growth curves, and sensor data from wearable devices. FDA has thus grown into an established field in statistics [14, 6, 9, 24]. Functional data intrinsically have infinite dimensions but are tractably handled by assuming they were generated from a smooth underlying process. The simplest form is a univariate random curve $\mathrm{x}(t)$ defined on a compact set over the real line $\mathbb{R}$. Without loss of generality, it is typically assumed that $\mathrm{x}(t)$ is a smooth stochastic process over the $[0, 1]$ interval domain. Let $x_i(t), i \in \{1, \ldots, n\}$ be a random sample of $\mathrm{x}(t)$, for instance, growth curves corresponding to different subjects in a population. In practice, each $x_i(t)$ can only be observed discretely over the time interval $[0, 1]$, and often only at a few irregularly spaced $n_i$ time points, namely $\{t_{ij}\}_{j=1}^{n_i}$. Moreover, the observed measurements are often noisy. Thus, the observed data are $\mathcal{D} = \{\{(t_{ij}, y_{ij})\}_{j=1}^{n_i}\}_{i=1}^{n}$, where $y_{ij} = x_i(t_{ij}) + \epsilon_{ij}$ and $\epsilon_{ij}$ are random noise effects. For many

---

[*]Correspondence: Ju-Sheng Hong <jsdhong@ucdavis.edu>.

Code available at: https://github.com/jshong071/SAND.

38th Conference on Neural Information Processing Systems (NeurIPS 2024).

applications, it is reasonable to assume that the noise is independent over $t$ and across subjects and follows a zero mean distribution with finite variance.

The objective of this paper is to recover the latent processes $x_i(t), i \in \{1, \ldots, n\}$, using the noisy/sparse observed data $\mathcal{D}$. We introduce an estimator that is effective for both finely and sparsely measured functional data (i.e. regardless of whether the number of measurements for the $i$-th subject, $n_i$, tends to infinity or is uniformly bounded by a constant). The latter case, $\sup n_i < \infty$, is common in longitudinal studies and called *sparse functional data* in FDA. Accurate recovery (imputation) of the underlying function from noisy/sparse observations can improve analyses of longitudinal data across many sciences.

The conventional approach to imputing longitudinal data is to employ a parametric model with mixed-effects, such as the linear mixed-effects model. This approach lacks the flexibility to capture the shapes of more complex trajectories, so a nonparametric method is preferred for imputing functional data. For instance, MICE [21] uses a chained equation approach, where the imputation of each missing value is conditioned on the imputed values of the other variables, resulting in more accurate and stable imputations compared to single imputation methods. However, the resulting imputed functions may not be smooth and to get stable imputations, many iterations are required if the percentage of missing values is high [17, 25].

A standard FDA approach that avoids this drawback is to first reduce each function to a finite but possibly growing dimensional vector and then reconstruct the trajectory of each subject using a particular set of basis functions associated with the dimension reduction approach [28]. Two widely used dimension reduction methods are the basis expansion approach [16, 3] and Functional Principal Component Analysis (FPCA) [2, 15, 18, 10, 27, 13, 12, 26]. Among these, FPCA provides the most parsimonious approach. The most popular FPCA approach is PACE [24, 4], which estimates the top principal component functions and the corresponding scores of each subject to reconstruct the latent process of this subject through the Karhunen-Loève decomposition.

All aforementioned approaches use the combined data from all subjects to conduct the imputation for each particular individual $x_i(\cdot)$. Another approach that works well for *densely observed* functional data is to apply a *smoothing* method to interpolate the data from an individual. While such smoothing does not borrow information across subjects, it can be effective when the number of measurements ($n_i$) is large for all subjects (proper smoothing can denoise the observations).

Recent advancements in deep learning for functional data imputation include techniques like *neural processes* [8] and GAIN [29]. Their applications to real data are however dwarfed by the widespread use of transformer networks [23]. Using a self-attention mechanism to discerningly attend to regional information across a domain, transformers offer a natural architecture for learning the right information to rely on in order to best estimate $x_i(t)$ at $t$. Specifically, *encoder-decoder* transformers use representations generated by the encoder to produce properly contextualized estimations through the decoder. We note that imputing sparsely observed functional data can be viewed as a sequence-to-sequence learning task, for which transformers should be well-suited. However, imputation via transformers fails to yield a naturally smooth function. In this paper, we introduce a small modification of transformers that addresses this issue and produces more accurate imputations.

**Summary of our contributions.** (1) We show empirically that a vanilla transformer is promising for functional data imputation – it outperforms PACE and neural processes but the imputations are not differentiable/smooth. (2) We introduce a novel network layer, SAND (**s**elf-**a**tte**n**tion on **d**erivatives), that can be learned end-to-end with the rest of a transformer architecture. Extending the attention module, our SAND layer guarantees smooth imputations and does not require any assumptions on the data distribution. (3) We prove that imputations from SAND are first-order differentiable (Theorem 1), and derive an analytical expression relating the number of hidden nodes and prediction error of a SAND network fit via empirical risk minimization (Theorem 2).

## 2 Related works

### 2.1 Sparse and intensively measured functional data

In most longitudinal studies involving human subjects, the number of measurements $n_i$ is often small (bounded). This type of data is called sparse functional data. When $n_i$ is large and grows with the sample size $n$ to infinity, such functional data are called intensively measured functional data. We

note that they are not time series data, which is a collection of data observed discretely on an equally spaced time grid for a single subject. There is no smooth latent process that generates this time series.

For intensively measured functional data, we can apply a kernel smoother to smooth the noisy observations individually for each subject to recover the underlying curve $x_i(t)$ and perform subsequent data analysis. Let $K(\cdot)$ be a kernel function (often a symmetric p.d.f). $x_i(t)$ is estimated by the weighted sum of observed data, $\{x_i(t_{ij})\}_{j=1}^{n_i}$, where weights are computed using kernel $K(\cdot)$: $\widehat{x}_i(t) = \sum_{j=0}^{n_i} K_h(t_{ij} - t)x_i(t_{ij})$ with $K_h(\cdot) = K(\cdot/h)/h$. The bandwidth $h$ that determines the size of the related neighborhood can be decided by generalized cross-validation [31].

It is not possible to consistently reconstruct the underlying curves for sparse functional data, due to the limited number of available observations per subject. It is however possible to consistently estimate the mean $\mu(t) = \mathbb{E}[\mathrm{x}(t)]$ and covariance function $G(s,t) = \mathrm{Cov}(\mathrm{x}(s), \mathrm{x}(t))$ from sparse functional data [27]. Once we have a consistent estimate of the covariance function, we can perform functional PCA and get the imputation of $x_i(t)$ as done in the popular PACE method [27]. Letting $\{(\lambda_k, \phi_k(\cdot))\}_{k=1}^{\infty}$ denote the eigen-components of the kernel operator defined by $G(s,t)$ (which can all be estimated consistently), PACE uses the Karhunen-Loéve series expansion of individual curves,

$$x_i(t) = \mu(t) + \sum_{k=1}^{\infty} \xi_{ik}\phi_k(t)$$

and replaces all unknowns with their estimates. To estimate the principal component scores $\xi_{ik}$, PACE uses the best linear predictor $\lambda_k\phi_{ik}^{\intercal}\Sigma_{\boldsymbol{y}_i}^{-1}(\boldsymbol{y}_i - \mu_i)$, where $\phi_{ik} = (\phi_k(t_{ij}))_{j=1}^{n_i}$, $\boldsymbol{y}_i = (y_{ij})_{j=1}^{n_i}$.

When $x_i(\cdot)$ is a Gaussian process, we expect the best linear predictor PACE to shine and this is reflected in the empirical results in Tables S9 and S10. These results further reveal that PACE continues to outperform existing imputation methods, such as MICE [22], CNP [7] and GAIN [29], when $x_i(\cdot)$ is not Gaussian. This outstanding performance underscores the status quo, that PACE is the state-of-the-art imputation method for functional data. However, PACE is a linear imputation method, which may restrict its ability to estimate a more complex data structure. In this work, we aim to replace this constraint with a more flexible model that uses our custom self-attention module.

## 2.2 Neural processes

The first study of deep learning for functional data imputation led to *neural processes* (NP) [7], which perform imputation by modeling the distribution of a *target* $x_i(t)$ conditioning on the observed *contexts* $\{(t_{ij}, y_{ij})\}$ via latent variables. From a set of observed data, the NP estimates a distribution over functions from which these data points could have been sampled. This estimation is done more flexibly than traditional *Gaussian Process* inference by utilizing a learned neural network. Variants of this approach include the *conditional neural process* (CNP) [7] and *attentive neural process* [11].

While the NP appears promising, we empirically find that straightforward sequence-to-sequence modeling via transformers yields more accurate functional imputations. Our proposed transformer extension learns to recover smooth trajectories from a fixed set of sparse observations in a single-shot. NP models are more complex and must learn their conditional distributions by iteratively sampling observations from the process. At each training iteration of an NP, one first samples a random function and then some observations from the function, repeating this procedure many times. In contrast, our method specifically aims to recover smooth trajectories from a given (fixed) set of observations, rather than relying on iterative sampling.

## 2.3 Self-attention

Transformer [23] comprises an encoder and a decoder, each consisting of stacks of attention and feed-forward layers. [30] highlight the roles of these layers: self-attention captures contextual mappings of the inputs and the feed-forward layer maps them to the desired output values. To provide a general overview of the self-attention module, we consider a generic real-valued input matrix $\boldsymbol{X}$ of size $q \times m$. Let $h_d$ be the number of hidden nodes and $H$ be the number of heads used in a self-attention module. For each head $h$, parameters $\boldsymbol{W}_O^{(h)} \in \mathbb{R}^{q \times h_d}$ and the key-value-query component $\boldsymbol{W}_V^{(h)}, \boldsymbol{W}_Q^{(h)}, \boldsymbol{W}_K^{(h)} \in \mathbb{R}^{h_d \times q}$ are learnable. Denote $\mathrm{softmax}(\boldsymbol{X})$ as applying the softmax operation

to columns of $\boldsymbol{X}$. A self-attention module defines a mapping from $\mathbb{R}^{q \times m}$ to $\mathbb{R}^{q \times m}$:

$$\text{Attn}(\boldsymbol{X}) = \boldsymbol{X} + \sum_{h=1}^{H} \left\{ \boldsymbol{W}_O^{(h)} \left( \boldsymbol{W}_V^{(h)} \boldsymbol{X} \right) \text{softmax} \left[ \left( \boldsymbol{W}_K^{(h)} \boldsymbol{X} \right)^\top \left( \boldsymbol{W}_Q^{(h)} \boldsymbol{X} \right) \Big/ \sqrt{h_d} \right] \right\}^\top . \quad (1)$$

The input $\boldsymbol{X}$ can be viewed as a $2 \times n_i$ matrix with the first and the second rows being $(y_{ij})_{j=1}^{n_i}$ and $(t_{ij})_{j=1}^{n_i}$, respectively. The $\text{softmax}$ operation generates a matrix with non-negative entries, where the columns sum up to 1. By multiplying softmax on the right of $\boldsymbol{X}$, we obtain a weighted sum of the columns of $\boldsymbol{X}$. This can be viewed as an interpolation method aiming to recover the underlying process $x_i(t)$ simultaneously for all subjects, by pooling all observed data together to estimate the weights $\boldsymbol{W}_K$ and $\boldsymbol{W}_Q$. This approach proves to be more efficient and distinct from individually interpolating each subject through a weighted sum of $\boldsymbol{X}$.

## 3 Method

As our objective is to smoothly impute curves, a transformer with ReLU activation function (RT) may not be the optimal choice as ReLU is not differentiable at 0. An alternative is to consider a GeLU-activated transformer (GT). Since GT outperforms RT in our benchmarks (Tables S2 and S3), we investigate GT and its variations and propose an architecture that equips GT with our SAND module in the rest of this section.

### 3.1 Functional imputation via the vanilla transformer network

Let $\mathcal{I} = \{0, \frac{1}{M-1}, \frac{2}{M-1}, \ldots, 1\}$ be a discretization of $[0, 1]$ where $M$ is the number of points, $\widetilde{T}_i$ be an imputation on an output grid $\mathcal{I}$ for subject $i$, and $\mathbb{A}$ be the training index set with $\|\mathbb{A}\|_0$ being the cardinality. A transformer network $f$ defines a mapping from $\mathbb{R}^{2 \times n_i}$ to $\mathbb{R}^M$ through $\widetilde{T}_i = f(\boldsymbol{S}_i)$ where $\boldsymbol{S}_i$ (source) is the input and $\widetilde{T}_i$ (target) is the output. During training, the mean squared error (MSE) loss between the observations $\boldsymbol{y}_i$ and the imputations on $\{t_{ij}\}_{j=1}^{n_i}$ is minimized. Let $\|\boldsymbol{x}\|_2$ denote the $\ell_2$ norm of $\boldsymbol{x}$. We use the same $\widetilde{T}_i$ as the imputation projected on $(t_{ij})_{j=1}^{n_i}$ and minimize

$$\text{MSE}_{\mathbb{A}} = \sum_{i \in \mathbb{A}} \|\widetilde{T}_i - \boldsymbol{y}_i\|_2^2 / \|\mathbb{A}\|_0. \quad (2)$$

During the imputation, we select a sufficiently large $M = 101$ to ensure that $\widetilde{T}$ accurately represents the imputation within the interval $[0, 1]$. However, due to the universal approximation capabilities of transformers [30] and the presence of noisy data during training, well-trained models can often produce zigzag-like imputations that mimic the noise in the training data. Meanwhile, it's essential to note that when these models encounter testing data, which is also observed with noise, they may continue to generate zigzag patterns if they have learned to replicate the noise patterns present in their training data, as shown in the first two plots of Figure 1. This can result in imputations that fail to capture the underlying, noise-free structure of the data.

A natural way to counter a non-smooth issue is to add a penalty. Let $\beta$ denote the collection of all parameters used in GT. By minimizing $\text{MSE}_{\mathbb{A}} + \lambda \|\beta\|_2^2$ where $\lambda$ is chosen by the validation dataset, we reduce the complexity of the model by shrinking the parameters towards zero. This results in a model that is less likely to fit the noise. However, it does not guarantee the smoothness of the target function (see the third plot of Figure 1). Alternatively, we explicitly penalize the output function by considering the objective function as $\text{MSE}_{\mathbb{A}} + \lambda \sum_{i \in \mathbb{A}} \int_0^1 [\widetilde{T}_i^{(2)}(t)]^2 \, dt / \|\mathbb{A}\|_0^2$. This objective function penalizes the curvature of the imputation and restricts its behavior. However, as illustrated in the fourth plot of Figure 1, this penalization approach fails to resolve the non-smooth issue.

Another approach is to apply statistical methods that ensure the differentiability of the imputations from transformers (either penalized or not). For instance, after the training of the network, we may apply PACE to estimate the eigen-components of the non-smooth outputs and use them to reconstruct the prediction output. A kernel smoothing can also individually smooth each of the outputs. However, transformers with these methods are not typically preferred due to their lack of end-to-end training. Related numerical results can be found in Section 5.1.

---

[2]In PyTorch, we use $\text{torch.diff}$ and $\text{torch.cumsum}$ to efficiently calculate the derivative and the integral.

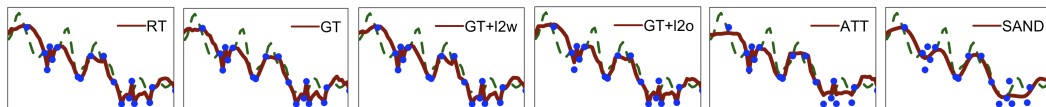

Figure 1: Close-ups of imputations generated by transformer variants (red curves) for testing data. Green lines are the true underlying functions from which observations (blue dots) are sampled with measurement errors. GT+l2w and GT+l2o are GTs regularized by the $\ell_2$ norm of its parameters and the curvature of imputations, respectively. SAND is our proposed method.

*Remark* 1. Here, we incorporate positional encoding for functional data and prepare it as input sequences for a transformer. Let $n_{\max} = \max_i n_i$. For each subject $i$, the locations $\boldsymbol{t}_i = (t_{ij})_{j=1}^{n_i}$ and values $\boldsymbol{y}_i = (y_{ij})_{j=1}^{n_i}$ are both padded with $(n_{\max} - n_i)$ zeros. To prevent attention modules from attending to the padding, we use mask vectors in the encoder and decoder components. To incorporate positional encoding, we define $\boldsymbol{E}_i \in \mathbb{R}^{p \times n_{\max}}$ as the encoding of $\boldsymbol{t}_i$, with $p$ being the encoding dimension. $(\boldsymbol{E}_i)_{jk}$ is computed as $\sin\left((M-1)t_{ik}/10000^{(j+1)/p}\right)$ for odd values of $j$, and $\cos\left((M-1)t_{ik}/10000^{j/p}\right)$ for even values of $j$. Finally, we stack $\boldsymbol{S}_i = [\boldsymbol{y}_i; \boldsymbol{t}_i; \boldsymbol{E}_i]$ as an input.

## 3.2 Self-attention on derivative

To overcome the lack of smoothness in imputations, we introduce a new module called **S**elf-**A**tte**N**tion on **D**erivative (SAND). SAND is stacked after GT, taking the imputations from GT as inputs, and outputting a smoother imputation in two steps: it first learns the derivative of the original imputation through a novel operator $\mathrm{Diff}(\cdot)$, which fathoms the derivative information and is inspired by the self-attention module in (1); then, it reconstructs a smooth version of the imputation via the numerical integral operator $\mathrm{Intg}(\cdot)$. We describe these two operators after introducing necessary notations.

Let $\widetilde{T}$ be an imputation on grid $\mathcal{I}$ with the first element $(\widetilde{T})_1$ and $\widetilde{T}_c = \widetilde{T} - (\widetilde{T})_1$ as $\widetilde{T}$ being shifted by $(\widetilde{T})_1$. Subsequently, let $\widetilde{\boldsymbol{T}}$ and $\widetilde{\boldsymbol{T}}_c$ be $(1+p)$-by-$M$ matrices where the first rows are $\widetilde{T}$ and $\widetilde{T}_c$, respectively, and the rest of rows are the $p$-dim positional encoding of $\mathcal{I}$. The symbol $\mathrm{cumsum}(\boldsymbol{x})$ is the vector of the cumulative summation of $\boldsymbol{x}$. Its $j$-th element $[\mathrm{cumsum}(\boldsymbol{x})]_j$ is $\sum_{k=1}^{j} x_k$. Let $W_O^{(h)}$, $\boldsymbol{W}_V^{(h)}$, $\boldsymbol{W}_K^{(h)}$ and $\boldsymbol{W}_Q^{(h)}$ be learnable parameters. The Diff and Intg operators are:

$$\mathrm{Diff}(\widetilde{T}) = \widetilde{D} = \sum_{h=1}^{H} W_O^{(h)} \left(\boldsymbol{W}_V^{(h)} \widetilde{\boldsymbol{T}}_c\right) \left[\left(\boldsymbol{W}_K^{(h)} \widetilde{\boldsymbol{T}}_c\right)^{\mathsf{T}} \left(\boldsymbol{W}_Q^{(h)} \widetilde{\boldsymbol{T}}_c\right) \Big/ \sqrt{h_d}\right], \qquad (3)$$

$$\mathrm{Intg}(\widetilde{D}) = \mathrm{cumsum}\left[\widetilde{D}/(M-1)\right]. \qquad (4)$$

Denote $\widehat{T}$ as the output from SAND. Then, SAND can be precisely summarized as follows:

$$\widehat{T} := \mathrm{SAND}(\widetilde{\boldsymbol{T}}) = (\widetilde{T})_1 + \mathrm{Intg}[\mathrm{Diff}(\widetilde{T})]. \qquad (5)$$

Figure 2 depicts the role of SAND: it takes an output $\widetilde{T}$ from a transformer, learns its derivative via Diff and Intg, and recovers the noise-free trajectory $\widehat{T}$ by minimizing the $\ell_2$ distance of $\widehat{T}$ and $\widetilde{T}$.

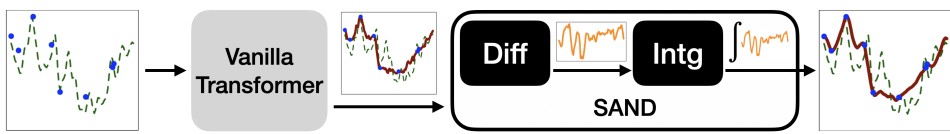

Figure 2: SAND's pipeline. Dashed lines are underlying processes, dots are observations sampled with errors, solid red curves are imputations, and solid orange curves are learned derivatives.

**Diff$(\cdot)$ operator.** The structure of $\mathrm{Diff}(\cdot)$ is similar to the attention module defined in (1) where the key-query product is scaled by $\sqrt{h_d}$ to prevent the product from growing with the hidden dimension. However, $\mathrm{Diff}(\cdot)$ removes the $\mathrm{softmax}$ operator on the key-query product because the product is used as weights while approximating derivatives. It should not be restricted to be positive only as the approximation of $f'(a) \approx \frac{f(b)-f(a)}{b-a}$, for any differentiable $f$ and provided that $b$ is close to $a$,

involves a weighted sum of $f(a)$ and $f(b)$ with both positive and negative weights. We remark that despite we don't have the observed derivative information to guide $\text{Diff}(\cdot)$ during the training process, we conduct a comparison using simulated data. Figure S1 compares the output from Diff with the derivative of the underlying process and indicates that $\text{Diff}(\cdot)$ adeptly captures the overall shape.

**Intg$(\cdot)$ operator.** The function $\text{Intg}(\cdot)$ approximates the integral $f(b) - f(a) = \int_a^b f'(x)\,dx$, where $f(\cdot)$ is differentiable. At a high level, because $\text{Diff}(\cdot)$ is a continuous operator and the integration of a continuous function is differentiable, the output of $\text{Intg}(\cdot)$ is continuously differentiable. This result is formally developed in Theorem 1. Furthermore, the $\text{Intg}(\cdot)$ operator allows SAND to impute the target process from any index $k_0$. We let the $k_0$th location serve the purpose of $(a, f(a))$ in the integral at the beginning of this paragraph, then the imputation from SAND at index $k_1$ is $\widetilde{T}_{k_0} + (-1)^{I(k_1 < k_0)} \sum_{k \in \mathcal{K}} (\widehat{D})_k / (M - 1)$ where $I(k_1 < k_0)$ is 1 if $k_1 < k_0$ and is 0 otherwise, and $\mathcal{K}$ is the set of indices between $k_0$ and $k_1$ (inclusive). Hence, SAND imputes trajectories in a bidirectional fashion. A similar approach, scheduled sampling, was explored by [1].

*Remark* 2. Conceptually, any machine learning model that handles vector inputs and outputs could be utilized to enhance the coarse imputation from GT, as there are numerous viable options. The attention module is adapted for two reasons:

**Computational Efficiency.** SAND is computationally efficient compared to other well-established vector-to-vector models like recurrent networks or convolutional networks. This efficiency is crucial in our choice, as highlighted by [23]. The attention module itself has demonstrated effectiveness across diverse applications, thanks in great part to its scalability.

**Achieving Smooth Outputs.** Our objective is to produce a smooth curve, which requires that its first derivative be continuous. Since most neural networks inherently model continuous functions, they are suited to this task. The first derivative $f'(a) \approx \frac{f(b) - f(a)}{b - a}$ involves a linear combination of function values. In constructing SAND, we chose the attention module because it inherently performs a weighted summation of its inputs, aligning well with our needs for modeling derivative.

## 4 Theoretical analysis

Theorem 1 demonstrates that the imputations from SAND are continuously differentiable. Theorem 2 establishes a crucial finding that highlights the improved accuracy of SAND: it can be viewed as a dimension reduction technique. Proofs of all results in the section are provided in the supplement. To begin Theorem 1, let $C^1(\mathcal{I})$ denote the set of continuously differentiable functions defined on $\mathcal{I}$.

**Theorem 1.** *Let $\widehat{T}$ denote the outputs from SAND. Then, $\widehat{T}$ is in $C^1([0, 1])$ when $M \to \infty$.*

Theorem 1 implies that the imputation from SAND is continuously differentiable. This is in contrast to GT, which is prone to overfitting and non-smoothness issues, as discussed in Section 3.1. The assumption that $M \to \infty$ is fairly mild as it could tend to $\infty$ at any rate, not depending on the sample size $n$. In simulations, we validate our choice of $M$ by reporting the total variation (TV) of the difference between an imputation and its underlying process. This measure is defined as the average of absolute differences between adjacent elements in the difference.

Since SAND aims at minimizing the $\ell_2$ distance between its output ($\widehat{T}$) and the imputation derived from GT (its input, $\widetilde{T}$), it is possible to drive its training error to zero by using a large number of hidden nodes ($h_d$). However, this needs to be avoided as the input $\widetilde{T}$ contains random noise. Instead, the goal for SAND should be to effectively disentangle the inherent smooth process from the random noise in $\widetilde{T}$. Therefore, a small training error $\epsilon$ should be incorporated in SAND. The choice of $\epsilon$ is tied to the number of hidden nodes $h_d$, which can be determined by the fraction of variance unexplained in FPCA [27]. We formally encapsulate this intuition in Theorem 2.

**Theorem 2.** *Define $\mathbb{A}$, $\widetilde{T}_i$, and $\widetilde{T}_{i,c}$ as in section 3. Suppose that $\{\lambda_j\}_{j=1}^M$ are the non-increasing eigenvalues of $\sum_i \widetilde{T}_{i,c}^\intercal \widetilde{T}_{i,c}$. For any $\epsilon \geq 0$, let $d_\epsilon = \min\{d \mid \sum_{j=1}^d \lambda_j / \sum_{j=1}^M \lambda_j \geq 1 - \epsilon\}$. Then, there exists a SAND satisfying: (1) $h_d \geq d_\epsilon$; and (2) the positional encoding dimension being $M$, so that for any imputation output $\widehat{T}_i$ from SAND, we have*

$$\frac{1}{\|\mathbb{A}\|_0} \sum_{i \in \mathbb{A}} \|d_i^{mask} \odot (\widetilde{T}_i - \widehat{T}_i)\|_2^2 \ \leq \ \frac{\epsilon}{\|\mathbb{A}\|_0} \sum_{i \in \mathbb{A}} \|\widetilde{T}_{i,c}\|_2^2$$

*where $d_i^{mask}$, the decoder masking for ith subject, is a vector of length $M$ whose jth element is 1 if $t = \frac{j}{M-1}$ is observed in subject $i$ and is 0 otherwise and $u \odot v$ represents the element-wise multiplication between vectors $u$ and $v$.*

The eigenvalues of $\sum_i \widetilde{T}_{i,c}^\intercal \widetilde{T}_{i,c}$ in Theorem 2 are used to determine the number of hidden nodes in SAND. As a consequence of Theorem 2, the following corollary shows the connection between SAND and dimension reduction techniques of FPCA.

**Corollary 1.** *Let SAND be defined as in Theorem 2 with $h_d = d_\epsilon$. Suppose that for all subjects, we have full observations on the output grid $\mathcal{I}$, i.e., all elements in $d_i^{mask}$ is 1. Then, SAND minimizes*

$$\min_{\mathbb{P}:rank(\mathbb{P})=d_\epsilon} \frac{1}{\|\mathbb{A}\|_0} \sum_{i\in\mathbb{A}} \|\widetilde{T}_{i,c} - \widetilde{T}_{i,c}\mathbb{P}\|_2^2 = \frac{\epsilon}{\|\mathbb{A}\|_0} \sum_{i\in\mathbb{A}} \|\widetilde{T}_{i,c}\|_2^2.$$

*Remark* 3. Corollary 1 shows that SAND can be seen as a dimension reduction technique by viewing $h_d$ and $\epsilon$ as the fraction of variance unexplained and the number of principal components needed to attain an $\epsilon$-error in FPCA, respectively. However, because of the presence of the decoder masking, $h_d$ and $\epsilon$ do not generally play the two roles. Still, Corollary 1 provides an intuitive insights of them.

## 5  Experiments

SAND is compared to seven baseline methods: PACE [27], FACE [26], mFPCA [13], MICE [22], CNP [7], GAIN [29], and 1DS (one-dimensional kernel smoothing to impute each subject separately). PACE estimates the Karhunen-Loéve presentation based on the function input. FACE imputes the missing data through a different covariance smoothing method from PACE. mFPCA uses a likelihood-based approach to estimate the functional principal components. MICE imputes the missing data based on the fully conditional specification, where each incomplete variable is imputed by a separate model. CNP parametrizes distribution over imputations giving a distributed representation of pairs of time points and functional values. GAIN imputes the missing data using generative adversarial nets. Due to the lack of available code, we did not include the methods of [16] and [10]. We discuss more on them along with [13] in the supplementary materials.

Each baseline involves the choice of hyperparameters. In PACE and mFPCA, we let the fraction of variance explained be 99%. FACE can select all hyperparameter automatically. In MICE, we impute 20 versions of data and take their mean, as suggested in [25]. In CNP, we implement [7] [link] using PyTorch to train the model for 50,000 epochs. The code for GAIN is provided in [29][link]. We train the model for a max of 10,000 epochs with 15 different random seed and take their mean. Finally, in 1DS, the bandwidth of the smoothing window is selected by cross-validation.

All transformers in Section 3 were trained for 5,000 epochs using a mini-batch of size 256 with a learning rate of $3 \times 10^{-4}$ and a 15% dropout rate. The penalty parameters are picked from a data-driven grid containing 8 points. In SAND, $h_d$ is set to $128$ and the dropout rate is 5%. See the code in the supplement for details of the model configuration. To quantify the uncertainty of the imputations, we add the pinball loss with the 10th and 90th target quantiles to the MSE in (2). Training takes 8 hours on one Nvidia A100 with 10,000 samples.

**Data splitting and evaluation of imputations.** The dataset is divided into three parts: the first 90% is for training, the last 5% for testing, and the remaining 5% for validation. The error between prediction and the underlying process on the testing data is reported for comparison. To evaluate the performance of each method, we report the MSE on testing data using (2) along with its standard error (SE). To validate Theorem 1 and our choice of $M$, we provide the mean and SE of the TV of the difference between any imputation $\widetilde{T}_i$ and the underlying process $x_i(\cdot)$.

### 5.1  Simulation studies

Through simulations, we show that transformer-based models outperform other imputation methods. Extensive simulation schemes varying in the number of basis functions, the number of points sampled in a curve, the size of random noises, and the independence of the sampled points are considered. Our simulations are purposely designed to probe the shorts of all baseline methods and neural networks.

**Data-generating process.** Let $x_i(t) = \sum_{k=1}^{K} [a_{ik} \sin(2\pi kt) + b_{ik} \cos(2\pi kt)]/k$ where $t \in [0, 1]$, $i \in \{1, \ldots, 10^4\}$ and $a_{ik}, b_{ik}$ follow a zero-mean distribution. The actual observed data of the ith

Table 1: MSE(SE) & TV(SE) on simulated data. Bold values indicate the top 2 performing methods.

| | $n_i = 30$ | | $n_i = 8$ to 12 | | $n_i = 3, 4, 5$ | |
|---|---|---|---|---|---|---|
| | MSE(SE) | TV(SE) | MSE(SE) | TV(SE) | MSE(SE) | TV(SE) |
| PACE | 189.9(4.3) | 187.1(2.0) | 450.0(15) | 201.9(2.1) | 795.5(33) | 209.5(2.2) |
| FACE | 284.6(8.8) | 198.9(2.1) | 488.2(16) | 204.5(2.2) | 807.1(32) | 209.5(2.2) |
| mFPCA | 224.7(5.8) | 192.0(2.1) | 480.3(16) | 204.0(2.2) | 787.1(31) | **209.3**(2.2) |
| MICE | 176.7(3.7) | 233.1(1.7) | 721.6(27) | 318.4(3.0) | 1416(57) | 332.7(2.8) |
| CNP | 290.4(11) | 198.9(2.0) | 551.3(21) | 207.6(2.1) | 920.3(52) | 211.9(2.2) |
| GAIN | 261.9(6.8) | 350.0(3.4) | 1454(52) | 413.1(5.1) | 1862(51) | 385.4(4.3) |
| 1DS | 262.9(6.0) | 273.8(2.4) | 735.3(22) | 305.7(3.7) | 1157(43) | 263.3(3.1) |
| GeLU-activated transformers with penalties | | | | | | |
| GT1 | 169.8(3.2) | 218.2(1.7) | 436.7(15) | 227.0(2.2) | 798.6(35) | 230.6(2.6) |
| GT1P | 169.0(3.5) | 179.9(2.0) | 425.3(14) | 199.4(2.1) | **777.4**(36) | 210.2(2.2) |
| GT1S | 169.8(3.2) | 218.1(1.7) | 436.7(15) | 227.0(2.2) | 796.5(35) | 227.6(2.6) |
| GT1T | **160.6**(3.2) | 171.9(1.9) | 421.8(14) | 196.8(2.1) | 783.4(34) | 211.8(2.2) |
| GT2 | 174.8(3.5) | 223.0(1.7) | 433.8(14) | 221.9(2.1) | 804.5(39) | 226.1(2.4) |
| GT2P | 179.9(3.9) | 182.1(2.0) | 422.6(13) | 199.4(2.1) | 788.9(39) | 210.0(2.2) |
| GT2S | 160.8(3.4) | **168.5**(1.5) | 427.7(13) | 208.8(2.1) | 804.5(39) | 226.1(2.4) |
| GT2T | 167.5(3.5) | 173.3(1.9) | **419.9**(13) | **197.1**(2.1) | 792.3(39) | 211.0(2.2) |
| GeLU-activated transformers with augmented modules | | | | | | |
| ATT | 185.1(3.8) | 220.0(1.7) | 446.9(14) | 220.6(2.1) | 852.0(42) | 224.0(2.5) |
| SAND | **146.5**(2.7) | **164.6**(1.8) | **410.9**(13) | **196.8**(2.0) | **758.1**(43) | **206.8**(2.2) |

curve is a discrete data $\{y_{ij}\}_{j=1}^{n_i}$, where $y_{ij} = x_i(t_{ij}) + \varepsilon_{ij}$ and $\varepsilon_{ij}$ follows $N(0, \sigma^2)$ independently. The output grid $[0, 1]$ is discretized in $\{j/100\}_{j=0}^{100}$ in all simulation. The experiment consists of 12 scenarios, varying in the distribution of the eigenvalues ($a_{ik}$ and $b_{ik}$), the number of basis functions $2K$, and the magnitude of $\sigma^2$. The full simulation is provided in the supplement (Table S1). In the main text, we only focus on the scenario when the eigenvalues follow an exponential distribution, which stimulates heavy-tailed processes, the number of basis functions is set to 40, the time points within any subject are sampled independently, and the signal-to-noise ratio (SNR) is set to 4. For a function input $f(t)$, we define $\text{SNR}(f)$ as $\int_{\mathcal{T}} f^2(t)\, dt / \sigma^2$. Finally, each scenario comprises three cases determined by $n_i$: either $n_i = 30$, $n_i = 8$ to 12, or $n_i = 3, 4, 5$.

**Transformer-based models.** In Table 1, GT1 and GT2 are GTs penalized with $\ell_2$ norm of parameters and the curvatures of the imputations, respectively, where the penalties are chosen by the validation dataset. For $i$ being 1 or 2, GT$i$P, GT$i$S and GT$i$T use PACE, kernel smoothing and trend filtering [19], respectively, to subsequently smooth the imputations obtained from GT$i$. GT variants are discussed in Section 3.1. Moreover, we investigate the impact of adding an extra self-attention module after the GT as it bears a similar structure to an interpolation method. We refer it as 'ATT' in our analysis.

**Results.** Table 1 presents the imputation MSEs and TVs (scaled by 1,000) along with their SEs for all 4 cases. Transformer-based models consistently perform the best across all benchmarks, indicating their efficacy for functional data imputation. We begin by examining GTs, which empirically outperform other baseline methods (Table 1). Despite this, imputations from penalized transformers still lack smoothness (Figure 1). Table 1 reinforces this observation by showing significantly higher TV in the imputations from GT1 and GT2 compared to imputations from PACE, FACE, and CNP. This lack of smoothness can be further mitigated by subsequently applying methods like PACE or individual smoothing to further refine GT imputations. Table 1 confirms that total variations in all GTs decrease, yet a closer look at the table reveals that PACE (GT1P and GT2P) can generally lower MSE, but falls short, particularly when $n_i$ is large, where PACE's posterior smoothing dramatically escalates the MSE. Individual smoothing (GT1S and GT2S) can improve imputations for dense functional data, yet has limited benefits for sparsely observed data. Although trend filtering methods (GT1T and GT2T) consistently reduce MSE and TV, their performance is ultimately surpassed by SAND. Lastly, using a self-attention module to boost GT doesn't yield positive results in our simulation. Figure 1 still displays non-smooth imputations, and MSEs increase across all scenarios, compared to GT variants.

Table 2: MSE(SE) and TV(SE) on real datasets. Bold values indicate the top 2 performing methods.

| | UK electricity | | | | | | Framingham study | |
| | $n_i = 30$ | | $n_i = 8$ to 12 | | $n_i = 3, 4, 5$ | | $n_i = 3$ to 11 | |
| | MSE(SE) | TV(SE) | MSE(SE) | TV(SE) | MSE(SE) | TV(SE) | MSE*(SE) | TV*(SE) |
|---|---|---|---|---|---|---|---|---|
| PACE | 12.8(1.8) | **19.0**(1.1) | **30.1**(4.5) | **21.1**(1.2) | **39.6**(5.2) | **21.9**(1.2) | 0.076(0.003) | **0.04**(0.002) |
| FACE | 15.8(2.1) | 21.3(1.2) | 32.5(5.4) | 22.6(1.2) | **39.6**(5.2) | 23.0(1.2) | 0.075(0.004) | **0.03**(0.002) |
| mFPCA | 16.4(2.0) | 22.2(1.2) | 34.8(4.9) | 23.2(1.2) | 41.7(5.4) | 23.3(1.2) | 0.057(0.003) | 0.04(0.002) |
| MICE | 20.4(2.2) | 67.8(3.3) | 40.0(4.5) | 65.4(2.8) | 75.4(8.6) | 71.4(1.5) | 0.199(0.011) | 0.32(0.013) |
| CNP | 23.0(3.5) | 21.4(1.2) | 31.5(4.3) | 22.1(1.2) | 47.9(7.1) | **22.7**(1.2) | 0.087(0.004) | **0.04**(0.002) |
| GAIN | 31.9(3.7) | 108(5.6) | 75.4(8.2) | 104(6.7) | 99.6(15) | 121(2.4) | 0.190(0.01) | 0.23(0.007) |
| 1DS | 17.3(2.2) | 19.4(1.1) | 50.0(7.0) | 22.8(1.3) | 105(18) | 44.1(2.7) | 0.075(0.004) | 0.05(0.003) |
| GT | **10.7**(1.8) | 20.6(1.1) | 31.2(3.3) | 23.2(1.3) | 42.6(5.6) | 38.5(2.5) | **0.004**(0.0002) | 0.13(0.005) |
| SAND | **10.0**(1.9) | **15.7**(0.9) | **26.7**(3.0) | **20.1**(1.2) | **38.3**(5.1) | 25.5(1.6) | **0.019**(0.001) | 0.09(0.004) |

In contrast, the SAND method we proposed consistently outperforms all other approaches, yielding the lowest MSEs and TVs across all scenarios. Table 1 reflects that SAND effectively reduces MSEs and TVs by an average of around 8% and 13%, respectively, compared to GT1. This is further substantiated by Figure 1, which visually demonstrates the smoothness of SAND-generated outputs, offering empirical support for the validity of Theorem 1. Within the domain of transformer-based models, our focus is not solely on reducing MSE but also on improving the smoothness of imputations. For this end, we recommend examining the SEs of TVs among different transformer-based models. Upon reviewing TV columns in Table 1, it becomes evident that SAND significantly outperforms all other transformer imputation methods. This highlights the effectiveness of SAND in enhancing the smoothness of imputations. Furthermore, SAND achieves smaller TVs compared to PACE and FACE, suggesting its capacity to capture underlying processes while maintaining low prediction errors.

## 5.2  Application to real datasets

**UK household electricity dataset.** We consider a dataset that records every half-hour the energy consumption of 5,494 randomly selected households in London. This results in a total of 96 observations from 11/13 (Wed) to 11/14 (Thurs) of 2013[3]. See [20] for details of the dataset. Our objective is to recover the underlying process of households' electricity usage where the underlying process is the Gaussian 6-hour moving average of the complete data. To align with our simulation designs, we randomly select the data into $n_i = 30$, $n_i = 8$ to 12, or $n_i = 3, 4, 5$ and we sample the time points in two ways - independently and dependently. Tables 2 and S11 report that SAND consistently produces the best performance across all settings. When the observed data contain errors, SAND excels in generating smooth trajectories, as illustrated in Figure S2. They also reveal a substantial reduction in TVs when comparing SAND to GT in all cases. This shows the effectiveness of SAND in enhancing the smoothness of imputations.

**Framingham heart study.** We apply SAND and the baseline methods to the longitudinal data from the Framingham Heart Study [5]. We focus on $n = 890$ subjects who were live at age 70. Our goal is to reconstruct BMI trajectories from ages 40 to 60 based on their irregular and sparse measurements within this range ($n_i = 3$ to 11). Here the MSE (marked by asterisks in Table 2) are measured based on the observed data. Figure S3 shows the imputations of two testing data from SAND and its main competitors (PACE, FACE, CNP and GT) and Table 2 highlights the substantial advantage of employing SAND, which reduces MSE by at least 75% for all competitors except for GT. Here GT has a lower MSE possibly due to overfitting as reflected in its TV and Figure S3. Regarding TV, which represents the total variation of the imputation in this analysis, PACE, FACE and CNP have smaller TVs than GT and SAND. Figure S3 indicates that they might undersmooth the trajectories. Considering both MSE and TV, SAND emerges as the most robust model across various scenarios.

## 5.3  Advantage of SAND: Enhancing Downstream Prediction Tasks

This section demonstrates better imputations lead to better downstream predictions in two scenarios: (1) predicting the average energy consumption on November 15, 2013 based on the imputed trajectories of energy consumption of the previous two days, (2) forecasting the average BMI between age 61

---

[3]During this period, the number of households that had complete observations was maximized.

and 65 in the Framingham Heart Study using the imputed trajectory of the same subject between age 40 and 60. To predict those outcomes, we leverage AdaFNN [28], a machine learning model capable of learning optimal basis representations between the imputations and the targets. Table 3 shows the result of SAND along with the three top competitors in imputation tasks, PACE, FACE and GT. It's evident that SAND consistently achieves the lowest MSE across all prediction tasks and the principle that better imputations yield more accurate downstream predictions. These findings position SAND as a promising tool for enhancing downstream prediction tasks across diverse domains.

Table 3: MSE(SE) on downstream tasks. Bold font marks the smallest MSE across methods.

|  | UK electricity | | | Framingham study |
|---|---|---|---|---|
|  | $n_i = 30$ | $n_i = 8$ to $12$ | $n_i = 3, 4, 5$ | $n_i = 3$ to $11$ |
| PACE | 5.77(0.9) | 11.8(2.2) | 15.5(2.9) | 2.23(0.7) |
| FACE | 7.23(1.4) | 10.8(2.1) | 13.7(2.3) | 2.10(0.6) |
| GT | 6.22(1.1) | 8.01(1.3) | **12.0**(1.9) | 2.31(0.6) |
| SAND | **5.19**(0.6) | **7.39**(1.1) | **12.0**(2.0) | **1.76**(0.5) |

## 6 Discussion

This paper introduces a novel SAND layer in a transformer for imputing sparse and noisy functional data. SAND, when paired with a vanilla transformer, constitutes an end-to-end learning mechanism that involves two main operations: Diff (3) and Intg (4). Initially, Diff takes the noisy imputed function from a vanilla transformer and concentrates on the sub-derivative of the imputation. Following this, Intg constructs a differentiable imputation, utilizing the fundamental theorem of calculus. SAND minimizes the $\ell_2$ distance between the imputed and the original transformer-estimated functions, eliminating the need to estimate the derivative of a noisy function.

From a theoretical perspective, we prove the smoothness of the imputation from SAND and explicitly determine the necessary number of hidden nodes for SAND to achieve an $\epsilon$ prediction error boundary on training data (up to a constant). Interestingly, the error boundary serves the role of the fraction of variance unexplained in FPCA literature. Traditional imputation methods, such as FPCA, employ a dimension-reduction strategy to learn trajectories via a finite-dimensional vector. Empirical benchmarks reveal that SAND surpasses FPCA when the curves aren't a Gaussian process with a low number of basis function. SAND effectively smooths noisy imputations from a vanilla transformer, and yields more accurate imputations than other techniques to encourage smoothness.

## 7 Acknowledgments

We would like to thank the reviewers for their constructive feedback. The research of Jane-Ling Wang and Ju-Sheng Hong was supported by NSF DMS 22-10891 and NSF 24-13924.

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
