# OpenReview forum: "SAND: Smooth imputation of sparse and noisy functional data with Transformer networks"
_NeurIPS.cc/2024/Conference — NeurIPS 2024 poster_

### Official Review · Reviewer_zTxC · 2024-07-10

**Soundness:** 4
**Presentation:** 4
**Contribution:** 4
**Rating:** 8
**Confidence:** 3

**Summary:**

The paper addresses the limitations of ordinary transformers for doing imputation of functional data over irregularly longitudinal data. The authors propose a novel new variant of transformer that takes derivatives into account, called SAND. Theoretically, it's shown that SAND with a certain number of hidden neurons can do functional imputation well. Then empirically, extensive experiments show that SAND perform much better than other methods.

**Strengths:**

1. Clear writing. Everything is explained quite well, and the flow of logic is smooth. It's immediately clear to me that the authors study an important problem and do well.
2. A complete story with theories and empirical results. The paper uses a lot of math, but it's used quite properly, because the mathematical comes naturally out of the structures of the problem itself, rather than some artificial assumptions. The theoretical analysis properly justifies the design the new transformer architecture and is very neat. Then there are extensive empirical studies that prove the usefulness of the new architecture.
3. The authors study a specific class of problems with great clarity. I'm so tired of papers claiming to have one method that improves things in great generality, which usually are not evaluated completely. It's good to see that we have steady progress in developing solid tools for specific problems of certain structures.

**Weaknesses:**

I'm not seeing effective weaknesses.

**Questions:**

No.

**Limitations:**

Yes, limitations are addressed adequately and no potential negative societal impact.

---

> ### Author Rebuttal · Authors · 2024-08-01
>
> We thank the reviewer for the valuable and positive feedback.

---

> > ### Comment · Reviewer_zTxC · 2024-08-07
> > **Thank you too**
> >
> > We thank the authors for their thanks.

---

### Official Review · Reviewer_pcLe · 2024-07-16

**Soundness:** 3
**Presentation:** 3
**Contribution:** 3
**Rating:** 6
**Confidence:** 3

**Summary:**

This paper studies the problem of how to perform imputation of the underlying function from noisy or sparse observations with functional data. In particular, the authors present "SAND" (Self-Attention on Derivatives), a variant of the transformer architecture, by introducing  $\mathrm{diff}(\cdot)$ (derivative) and $\mathrm{Intg}(\cdot)$ (integral) operators to the standard transformer, to address the imputation of sparse and noisy functional data. The authors provide theoretical guarantees for the proposed SAND method as well as empirical evaluations across various datasets demonstrate that SAND outperforms existing methods.

**Strengths:**

- The proposed method with the two ingredients, $\mathrm{diff}(\cdot)$ and $\mathrm{Intg}(\cdot)$, in the modeling part are well-motivated for tackling the smoothness issue.

- Empirically the paper demonstrates that the proposed SAND transformer can achieve strong performance on both synthetic and real-world datasets.

**Weaknesses:**

- The theoretical results seem only justify the proposed architecture can approximate the FPCA, which do not provide insights on why this new transformer variant is better than the standard transformer for solving the imputation problem.
- The design of the $\mathrm{diff}(\cdot)$ operator is not very intuitive, since the operator itself does not align well with the derivative operator.
- I would suggest add a simple baseline (standard transformer with some smoothing post-processing) for comparison. For example, applying certain smoothing technical on top of the output of standard transformers, e.g., locally adaptive regression splines (LARS, Mammen & van de Geer, 1997), trend filtering (Tibshirani, 2014).

[Mammen & van de Geer, 1997] Enno Mammen and Sara van de Geer. Locally adaptive regression splines. The Annals of Statistics, 25(1):387–413, 1997.
[Tibshirani, 2014] Ryan J Tibshirani. Adaptive piecewise polynomial estimation via trend filtering. The Annals of Statistics, 42(1):285–323, 2014.

I am happy to raise my score if some of the questions/weaknesses are addressed during rebuttal and discussion.

**Questions:**

- The notation in Section 3.1 and Section 3.2 is a bit unclear to me. For example, the bolded tilde $T$ and $T_c$ are $(1+p)\times M$ matrices, I guess there are positional embeddings inside bolded tilde $T$ and $T_c$. However, these are not formally defined. I would suggest add a notation paragraph to clarify the notation somewhere (could be in appendix in the main body has limited space).
- In Line 184, why subtracting the first element (or why constructing the tilde $T_c$)? I guess it is because the $\mathrm{diff}(\cdot)$ and $\mathrm{Intg}(\cdot)$ operators. Would removing this shift term affect the performance of SAND?

---

> ### Author Rebuttal · Authors · 2024-08-06
>
> We thank the reviewer for the valuable and positive feedback, which we have used to strengthen our paper.  Please see our responses below.
>
> ### [Weakness 1]
> **Answer**: Thank you for the comment. We acknowledge that most of our theorems focus on SAND. However, our theorem does reveal one reason that SAND is better than the standard Transformer:  See Corollary 1 and lines 223 to 226 of the manuscript, which explain how SAND effectively separates the smooth underlying process from random noise by limiting hidden nodes in the network. The standard Transformer is incapable of doing so due to its universal approximation property and the existence of noise in the training data (lines 149 to 154 in the manuscript). When the standard Transformer encounters noisy testing data, they often continue producing similar zigzag patterns, as they have learned to replicate the noise from the training phase.
>
> ### [Weakness 2]
> **Answer**: The traditional numerical derivative operator takes the form $f^\prime(a) \approx \frac{1}{b - a}[f(b) - f(a)]$, a linear combination of the inputs $f(a)$ and $f(b)$ with fixed weights (line 195). Our $\rm{diff}(\cdot)$ operator similarly calculates a weighted sum of these inputs, with weights learned by the network.
> Our decision to design the $\rm{diff}(\cdot)$ operator to resemble the attention module was driven by two main considerations:
> - **Cohesion with the Transformer Architecture**: Our SAND module is a simple  augmentation of the standard Transformer. SAND modifies the conventional attention module by omitting the softmax operation, thereby aligning its computational complexity with that of the original attention mechanism, as detailed in Table 1 of Vaswani et al. (2017). Compared to recurrent and convolutional networks (other architectural techniques for achieving smoothing), SAND enjoys major computational advantages (Vaswani et al., 2017).
> - **Leveraging Proven Mechanisms:** The attention mechanism has demonstrated effectiveness across numerous tasks. By minimally adapting this component, we aim to preserve this versatility in our operator. The $\rm{diff}(\cdot)$ operator retains the core benefits of attention while extending its application to approximate derivatives and enforce smoothness.
>
> ### [Weakness 3]
> **Answer**: We have indeed considered the approach of post-processing imputations from a standard Transformer. As discussed in lines 164 to 168 and in Table 1 under the rows labeled “GT1P”, “GT1S”, “GT2P”, and “GT2S”, we applied both PACE and kernel smoothing as post-processing methods to imputations from a standard Transformer model.
> The effectiveness of these methods is discussed in lines 302 and 307, where we note that while the total variation of the estimated function generally decreases, the improvement in MSE is marginal, and in some cases, the MSE actually increases after post-processing. These findings underscore the limitations of simply adding some smoothing post-processing to standard Transformers, regardless of which smoothing technique is used.
> We chose PACE as a popular imputation method for noisy functional data, and we chose kernel smoothing due to its simplicity and historical prominence as the canonical  smoothing method. Following your suggestion, we have also applied trend filtering to the output of standard Transformer, by using  `trendfilter` from the package `genlasso` in `R`. The results are in the PDF file attached to our global rebuttal. Although the improvement is greater than using PACE or kernel smoothing as post-processing, SAND remains the best overall method. Our revised paper incorporates and discusses these results, as well as accounting for the other valuable feedback.
>
> ### [Question 1]
> **Answer**: Yes, there are positional embeddings inside bolded $\tilde{T}$ and $\tilde{T}_c$. The notation is defined in lines 185 and 186: bolded $\tilde{T}$ (or $\tilde{T}_c$) are matrices with the first row being $T$ (or $T_c$), and the rest of $p$ rows are the positional encoding of the output grid. Thank you for the suggestion. Due to the limited space in the main text, we will add a notation paragraph to in the appendix.
>
> ### [Question 2]
> **Answer**: The subtraction of the first element serves multiple purposes:
> - **Geometrical Justification**: This adjustment ensures that two curves, which may have identical shapes but different intercepts (the first element in $T$), are treated equivalently by the $\rm{diff}(\cdot)$ operator. By subtracting the first element, we effectively remove the absolute positioning, focusing solely on the shape of the curve.
> - **Mathematical Rationale**: Our approach is inspired by the first fundamental theorem of calculus, which expresses a function as $f(b) = f(a) + \int_a^b f’(x),dx$ (refer to line 201 in the main text). In the context of SAND, $f(a)$ represents the first element of $T$, serving as the initial value. The $\rm{Intg}$ function acts as "$\int$", and $\rm{diff}$ approximates the derivative. Subtracting the first element aligns with this conceptual framework, where $f(a)$ is reintegrated post-differentiation to reconstruct the curve accurately (see Equation 5 between lines 189 and 190).
> - **Algorithmic Efficiency**: Subtraction normalizes the input data, enhancing the uniformity of the inputs. This normalization can lead to faster convergence of the $\rm{diff}(\cdot)$ operator’s parameters during training.
> Omitting the shift could potentially require more training epochs for SAND to achieve similar convergence, due to the variability in initial values across data samples.

---

> > ### Author Response · Authors · 2024-08-11
> >
> > Dear Reviewer,
> >
> > Thank you again for your valuable feedback on our paper.  We have addressed your questions to the best of our ability and have revised our paper based on your suggestions (including additional results and more clearly written motivations). The revised paper is much stronger thanks to your feedback.
> >
> > Please let us know If you have any follow-up questions/concerns, and we will address them before the discussion periods ends on Aug 13.  We value your insights and are eager to further improve our paper based on your follow-up thoughts.

---

> > ### Comment · Reviewer_pcLe · 2024-08-13
> > **Response**
> >
> > I would like to thank the authors for their response. The added new experimental results further demonstrate the benefits of the new architecture. I have increased my score.

---

### Official Review · Reviewer_92dE · 2024-07-19

**Soundness:** 2
**Presentation:** 3
**Contribution:** 3
**Rating:** 5
**Confidence:** 5

**Summary:**

This paper proposes a new class of transformers for sparse and noisy functional data. In particular, a new module, namely self-attention on derivatives (SAND), is incorporated vanilla transformers to model the sub-derivative of the imputed curve, thereby promoting smoothness. The authors also theoretically prove the number of hidden nodes needed by the SAND transformer to achieve a certain prediction error bound for functional imputation. Empirical results are provided to justify the advantages of the proposed model.

**Strengths:**

1. The proposed method, i.e., SAND, is interesting.

2. Theoretical properties of SAND is well-studied.

3. The paper is well-written with illustrative figures.

**Weaknesses:**

1. Experiments of larger-scale benchmarks are needed to justify the advantages of SAND.

2. Efficiency analysis of SAND is missing.

3. The advantages of using attention in SAND is not clearly discussed in the paper.

**Questions:**

1. Can you please elaborate more on the use of attention in SAND? Why do we need attention there?

After Rebuttal

I will increase my score from 4 to 5.

**Limitations:**

The authors have adequately addressed the limitations.

---

> ### Author Rebuttal · Authors · 2024-08-06
>
> We thank the reviewer for the valuable and positive feedback, which we have used to strengthen our paper.  Please see our responses to the weakness below. Our responses to the questions are listed in the global author response.
>
> ### [Weakness 1]
> **Answer**:
> Thank you for highlighting this aspect. We have not scaled up the simulation for three main reasons:
> - **Simulation Constraints**: As detailed in line 267, each simulated case requires one day for processing on an Nvidia GeForce GTX 1080 Ti. With a total of 88 cases outlined across Tables S2-S10 in the supplementary material, completing all simulations under these conditions requires nearly three months. Even with early-stopping techniques, it still takes up to two months to complete the simulation. These time constraints make larger simulations impractical under our compute budget.
> - **Relevance to Real-World Data**: The sample size of 10,000 in our simulations is substantial, particularly when compared to real datasets we have used, such as the UK electricity dataset with 5,600 samples and the Framingham Heart Study with 870 samples. Simulating data at a scale comparable to real-world datasets ensures that our findings are both realistic and applicable.
> - **Effectiveness of SAND at current scale**: Despite perceptions that 10,000 samples may not seem extensive, our simulations demonstrate a significant advantage of SAND over nine competitors. This indicates robust performance even at the current scale, supporting the effectiveness of SAND on datasets of similar size to popular real-world functional imputation tasks.
>
> ### [Weakness 2]
> **Answer**: Thank you for highlighting the importance of providing a detailed efficiency analysis. SAND modifies the conventional attention module by omitting the softmax operation, thereby aligning its computational complexity with that of the original attention mechanism, as detailed in Table 1 of Vaswani et al. (2017).  Let $m$ denote the number of observations in a subject, $h_d$ be the representation dimension, and $k$ be the kernel size of convolutions. We summarize the computational efficiency of SAND as follows:
> - **Compared to Recurrent Modules**: SAND requires $O(1)$ sequential operations, regardless of sequence length, facilitating faster computation and better suitability for parallel processing. In contrast, recurrent layers inherently require $O(m)$ sequential operations due to their dependency on previous outputs for current computations, which significantly slows down processing and limits scalability.
> - **Compared to Convolutional Modules**: The computational complexity of SAND is O(m^2\cdot h_d), whereas for convolutional modules, it’s $O(k\cdot m\cdot h_d^2)$. In our simulation and data applications, the number of observations per subject $m$ is at most 30 while the representation dimension is 128. Given these parameters, convolutional operations become computationally intensive, leading to slower performance compared to SAND.
> SAND, analogous to a single attention module, introduces minimal additional computational overhead compared to standard transformers, which typically comprise hundreds of alternating layers of attention and feed-forward modules. We acknowledge the significance of this aspect and our revised paper clearly articulates the computational complexity of SAND as well as its computational advantages.
>
> Regarding SAND’s prediction efficiency (in the sense of statistical estimation): Our evaluation of SAND’s prediction efficiency, measured through relative MSE, is detailed in Table 1. The results indicate that SAND is approximately 9% more efficient than a standard transformer and 11% more efficient than the best non-transformer benchmark, which is PACE. This demonstrates not only SAND’s computational advantages but also its superior accuracy in predictive tasks.
>
> This multi-dimensional approach to assessing efficiency—encompassing computational, estimation, and prediction aspects—provides a robust evaluation of SAND’s performance across various metrics.
>
> ### [Weakness 3]
> Thank you for pointing out the need for a clearer discussion on the benefits of incorporating the attention mechanism in SAND. Here are the key advantages:
> - Performance Enhancement: In section 5.1, our extensive experiments demonstrate that SAND significantly improves imputation accuracy, as evidenced by reductions in mean square error and total variation compared to standard methods. This performance boost underscores the effectiveness of the attention-based approach in handling functional data.
> - Addressing Non-linearity: Traditional post-processing techniques (see Table 1 in the manuscript) in functional data analysis, such as PACE, are typically linear and may fail to capture complex, non-linear features in the data (lines 101 to 102 in the manuscript). SAND effectively bridges this gap by utilizing the non-linear modeling capabilities of the attention mechanism, allowing for a more nuanced and powerful analysis of functional data.
> - Computational Efficiency: Leveraging the attention module within SAND capitalizes on the computational efficiencies discussed in response to [Weakness 2] and as detailed by Vaswani et al. (2017).
> Our revised paper more clearly lists these advantages of utilizing the attention module in SAND.
>
> ### [Question 1]
> **Answer**: Please see our response to the integrated question in the global author rebuttal for this question.

---

> > ### Author Response · Authors · 2024-08-11
> >
> > Dear Reviewer,
> >
> > Thank you again for your valuable feedback on our paper.  We have addressed your questions to the best of our ability and have revised our paper based on your suggestions (including additional results and more clearly written motivations). The revised paper is much stronger thanks to your feedback.
> >
> > Please let us know If you have any follow-up questions/concerns, and we will address them before the discussion periods ends on Aug 13.  We value your insights and are eager to further improve our paper based on your follow-up thoughts. In particular, our revision now more clearly motivates the use of attention. Any feedback on this updated explanation would be appreciated!

---

> > ### Comment · Reviewer_92dE · 2024-08-13
> >
> > Thank the authors for the detailed response. Can you please provide SAND's runtime and memory usage in comparison with the baseline? I know asking for additional experimental results less than 1 day before the deadline is inappropriate, but those results should be easy and quick to acquire. I need to know SAND's runtime and memory usage compared with the baseline to decide to raise my score or not.

---

> > > ### Author Response · Authors · 2024-08-14
> > >
> > > Thank you for your request for additional details concerning the runtime and memory usage of SAND compared to baseline methods. Below, you’ll find a table comparing SAND with eight other baseline models.
> > >
> > > The table includes each model’s number of parameters, memory usage, runtime, and the number of epochs required for convergence. It highlights that SAND, while enhancing the capabilities of a standard transformer, requires only minimally more memory and runtime. Notably, while models such as CNP and GAIN have smaller model sizes, their requirement for a significantly higher number of epochs (as per the epoch suggestions on their respective GitHub repositories) to achieve convergence means that their total runtime may not be shorter than that of transformer-based models. This aspect is crucial in understanding the efficiency and practical applicability of SAND in real-world scenarios.
> > >
> > > |                      | Use GPU? | num of params | memory usage* |      Runtime     | num of epochs |
> > > |:--------------------:|:--------:|:-------------:|:------------:|:----------------:|:-------------:|
> > > |         PACE         |     X    |       NA      |    2.84GB    |      41 secs     |       NA      |
> > > |         FACE         |     X    |       NA      |    11.20GB   |     1218 secs    |       NA      |
> > > |         mFPCA        |     X    |       NA      |    4.37GB    |     635 secs     |       NA      |
> > > |         MICE         |     X    |       NA      |     434MB    |     1800 secs    |       NA      |
> > > |          1DS         |     X    |       NA      |     <10MB    |      5 secs      |       NA      |
> > > |          CNP         |     V    |      83K      |     310MB    |  1.8 secs/epochs |     50,000    |
> > > |         GAIN         |     V    |      164K     |     460MB    |  1.2 secs/epochs |    150,000    |
> > > | Standard Transformer |     V    |      930K     |    3.61GB    | 15.6 secs/epochs |     5,000     |
> > > |         SAND         |     V    |      996K     |    3.97GB    | 16.4 secs/epochs |     5,000     |
> > > *  Memory usage values represent peak usage during the training phase.

---

> > > > ### Comment · Reviewer_92dE · 2024-08-14
> > > >
> > > > Thanks to the authors for providing the runtime and memory usage that I requested. I have no further questions and will increase my score from 4 to 5.

---

### Author Rebuttal · Authors · 2024-08-06

We are grateful for the detailed feedback from all reviewers, which has significantly contributed to refining our manuscript. We would like to address a specific concern raised by two reviewers (**92dE** and **pcLe**) regarding the use of attention in SAND's $\rm{diff}(\cdot)$ operator and the design of it. These concerns relate to fundamental aspects of our methodology and its innovative application in functional data imputation.

### [Integrated Question] Can you elaborate on the conceptual foundations and decision-making process behind the integration of the attention mechanism and the design of the $\rm{diff}(\cdot)$ operator in SAND?
**Answer**: We arrived at our decision to use the attention module in SAND as follows:  We initially started with a coarse imputation from a standard Transformer and aimed to refine its poor performance. One intuitive method could involve adding penalties to the standard transformer to constrain its outputs or applying posterior smoothing techniques to enhance the coarse imputation. We discuss the limitations of these approaches in lines 156 to 169 and benchmark their empirical performance in section 5.1.
An alternative approach is to apply a “patch” to the standard transformer such that the resulting architecture outputs a smooth imputation, a process depicted in Figure 2. Conceptually, we could utilize any machine learning model that handles vector inputs and outputs to achieve this, as there are numerous viable options.
We adapt the attention module as our patch in SAND for several reasons:
- Computational Efficiency: SAND is computationally efficient compared to other well-established vector-to-vector models like recurrent networks or convolutional networks. This efficiency is crucial in our choice, as highlighted by Vaswani et al. (2017). The attention module itself has demonstrated effectiveness across diverse applications, thanks in great part to its scalability.
- Achieving Smooth Outputs: Our objective is to produce a smooth curve, which requires that its first derivative be continuous. Since most neural networks inherently model continuous functions, they are suited to this task. The first derivative $f^\prime(a) \approx \frac{1}{b - a}[f(b) – f(a)]$ involves a linear combination of function values. In constructing SAND, we chose the attention module because it inherently performs a weighted summation of its inputs, aligning well with our needs for modeling derivatives (as detailed in Equation 1 and lines 130 to 134 in the manuscript).

These considerations guided our decision to employ the attention mechanism in SAND, which enjoys favorable theoretical properties and empirical performance.

We provide individual responses below to address the remaining concerns from each reviewer to improve clarity of missing details and to provide additional discussion that strengthen our paper. We thank all reviewers’ for their time and efforts! We hope our responses have persuasively addressed all remaining concerns. Please don’t hesitate to let us know of any additional comments or feedback on improvement.

Note that we include all additional experimental results in the one-page pdf submitted along with this global rebuttal response.

---

### Decision · Program_Chairs · 2024-09-25

**Decision:**

Accept (poster)

**Comment:**

All reviewers were in agreement that this paper should be accepted, as it develops a well motivated method and especially due to the strong experimental results. However I strongly urge the authors to address the concerns that came up during the process. While I cannot summarize everything that came up, here is one issue that the authors should address: Larger scale experiments should be done (the Nvidia GeForce GTX 1080 Ti that the authors refer to is by now vastly outdated--- it was released in 2017, before the Attention mechanism!).